# Plant growth promoting endophyte *Burkholderia contaminans* NZ antagonizes phytopathogen *Macrophomina phaseolina* through melanin synthesis and pyrrolnitrin inhibition

Nazia R. Zaman[1]☯, Umar F. Chowdhury[1]☯, Rifath N. Reza[1], Farhana T. Chowdhury[1], Mrinmoy Sarker[2], Muhammad M. Hossain[2], Md. Ahedul Akbor[3], Al Amin[1], Mohammad Riazul Islam[1]*, Haseena Khan[1]*

1 Molecular Biology Laboratory, Department of Biochemistry and Molecular Biology, Faculty Biological Sciences, University of Dhaka, Dhaka, Bangladesh, 2 NSU Genome Research Institute (NGRI), Department of Biochemistry and Microbiology, North South University, Dhaka, Bangladesh, 3 Institute of National Analytical Research and Service (INARS), Bangladesh Council of Scientific and Industrial Research (BCSIR), Dhaka, Bangladesh

☯ These authors contributed equally to this work.
* mriazulislam@du.ac.bd (MRI); haseena@du.ac.bd (HK)

## Abstract

The endophytic bacterium *Burkholderia contaminans* NZ was isolated from jute, which is an important fiber-producing plant. This bacterium exhibits significant growth promotion activity in *in vivo* pot experiments, and like other plant growth-promoting (PGP) bacteria fixes nitrogen, produces indole acetic acid (IAA), siderophore, and 1-aminocyclopropane-1-carboxylate (ACC) deaminase activity. *B. contaminans* NZ is considered to exert a promising growth inhibitory effect on *Macrophomina phaseolina*, a phytopathogen responsible for infecting hundreds of crops worldwide. This study aimed to identify the possibility of *B. contaminans* NZ as a safe biocontrol agent and assess its effectiveness in suppressing phytopathogenic fungi, especially *M. phaseolina*. Co-culture of *M. phaseolina* with *B. contaminans* NZ on both solid and liquid media revealed appreciable growth suppression of *M. phaseolina* and its chromogenic aberration in liquid culture. Genome mining of *B. contaminans* NZ using NaPDoS and antiSMASH revealed gene clusters that displayed 100% similarity for cytotoxic and antifungal substances, such as pyrrolnitrin. GC-MS analysis of *B. contaminans* NZ culture extracts revealed various bioactive compounds, including catechol; 9,10-dihydro-12'-hydroxy-2'-methyl-5'-(phenylmethyl)- ergotaman 3',6',18-trione; 2,3-dihydro-3,5- dihydroxy-6-methyl-4H-pyran-4-one; 1-(1,6-Dioxooctadecyl)- pyrrolidine; 9-Octadecenamide; and 2-methoxy- phenol. These compounds reportedly exhibit tyrosinase inhibitory, antifungal, and antibiotic activities. Using a more targeted approach, an RP-HPLC purified fraction was analyzed by LC-MS, confirming the existence of pyrrolnitrin in the *B. contaminans* NZ extract. Secondary metabolites, such as catechol and ergotaman, have been predicted to inhibit melanin synthesis in *M. phaseolina*. Thus, *B. contaminans* NZ appears to inhibit

**Data Availability Statement:** All relevant data are within the manuscript and its Supporting Information files.

**Funding:** This project is funded by Higher Education Quality Enhancement Project (HEQEP) (Grant number: CP-3250), a World Bank financed development project. The funders had no role in study design, data collection and analysis, decision to publish, or preparation of the manuscript.

**Competing interests:** The authors have declared that no competing interests exist.

phytopathogens by apparently impairing melanin synthesis and other potential biochemical pathways, exhibiting considerable fungistatic activity.

## Introduction

The continued use of expensive chemical fertilizers and fungicides have adversely affected human health and negatively impacted the environment; hence, the safe use of microorganisms that improve soil fertility, enhance plant growth and limit the growth of phytopathogenic fungi has been receiving immense attention from researchers [1]. Endophytes, mainly bacteria and fungi that spend at least a part of their lifespan within plant tissues without negatively affecting their hosts [2], have been intensely studied for years in terms of their diversity, metagenomics, combinatorial biosynthesis, plant growth promotion, biocontrol, bioremediation, etc. [3]. Plant-associated bacteria that aggressively colonize as symbiotic partners in the plant rhizosphere and roots with beneficial effects on plant growth are considered plant growth-promoting rhizobacteria (PGPR) [4]. In contrast, endophytic bacteria colonize the apoplasm or symplasm of the internal tissues of plants [5]. In general, with respect to growth promotion and protection against microbial infection, the beneficial effects of endophytes are considerably greater than those of several rhizobacteria [6].

The endophytes are grouped into three major clusters based on their different mechanisms of action. These contain: (i) biofertilization, which includes siderophore production, atmospheric nitrogen fixation, phosphate solubilization, and exopolysaccharides production; (ii) phyto-stimulation, which includes production of indole acetic acid, gibberellin, cytokinin, and ethylene; and (iii) phyto-biocontrol, which includes competing for iron, nutrients, and space; production of antibiotics, lytic enzymes, volatile compounds; and induction of systemic resistance [7]. Many studies have emphasized the ability of these microorganisms to promote and protect plant growth through additive/synergistic effects [1]; among them, *Streptomyces* spp. [8], *B. subtilis* [9], *Pseudomonas parafulva*, and *Pantoea agglomerans* [10] are a few examples.

Jute (*Corchorus olitorius* var. O-4) is an important natural fiber-producing cash crop in Southeast Asia [11]. Throughout its life cycle, jute is confronted by the destructive necrotrophic fungal pathogen *Macrophomina phaseolina* (Tassi) Goid., which causes charcoal rot disease in more than 500 plant species from approximately 100 families [12].

While screening for jute endophytes with potential inhibitory effects against *M. phaseolina*, we isolated an endophytic bacterium, *Burkholderia contaminans* NZ from the seed, which demonstrated promising antifungal activity [13]. This bacterium contained genes that are characteristic of PGPR, such as siderophore and ACC deaminase activities, phytohormone auxin (IAA) production, and nitrogen-fixing abilities, thereby promoting plant growth.

The genus *Burkholderia*, which belongs to the subphylum of β-proteobacteria, currently consists of 90 validly named species and several uncultivated candidate species [14]. They are ubiquitous organisms with high genetic versatility and adaptability, and are widespread in water, soil, plants, and animals, including humans [15].

However, the *Burkholderia* genus also comprises certain pathogenic species that threaten plant, animal, and human health [16]. *Burkholderia contaminans* is a member of the *Burkholderia cepacia complex* (Bcc) [17] which includes several closely related *Burkholderia* species. These opportunistic pathogens frequently colonize the lungs of cystic fibrosis (CF) and immune-compromised patients [18]. Furthermore, Bcc is the most frequently isolated clinical pathogen among *Burkholderia* spp., followed by *B. mallei* and *B. pseudomallei* [19]. Therefore,

the environmental release of *Burkholderia* species as a biocontrol agent in agriculture is severely debated due to difficulties in distinguishing plant growth-promoting and pathogenic bacteria [20]. Recent developments in *Burkholderia* taxonomy and molecular analysis may help in answering questions concerning differences between the pathogenic and ecological properties of *Burkholderia* species in an attempt to reconsider the possibility of using selected strains for biocontrol [21].

The present study aimed to determine the safety of *B. contaminans* NZ as a biocontrol agent and its effectiveness against phytopathogens, especially *M. phaseolina*. The entire genome sequence was analyzed in this study to unravel information regarding the safety of *B. contaminans* NZ as a PGPR and biocontrol agent based on their siderophores and secondary metabolites producing capacities, and the absence of virulence genes. *In vitro* data also revealed the attributes of growth promotion. Moreover, Gas chromatography–mass spectrometry (GC-MS) and liquid chromatography–mass spectrometry (LC-MS) analyses identified several potent antimicrobial compounds and antagonists of melanin synthesis, and the electron transport system produced by the bacterium in the presence of *M. phaseolina*. These compounds possibly antagonize fungal growth through different modes of action.

## Materials and methods

### Bacterial, fungal strains and plant materials

The phytopathogenic fungi *M. phaseolina* was obtained from the Bangladesh Jute Research Institute (BJRI), Dhaka. *Nigrospora sphaerica*, *Xylaria* spp., *Aspergillus fumigatus*, *Aspergillus niger*, and *Penicillium oxalicum* were collected from the Molecular Biology Lab, Department of Biochemistry and Molecular Biology, University of Dhaka. *Rhizoctonia solani* was obtained from the Bangladesh Agricultural University, Mymensingh, Bangladesh. All the fungi were grown and maintained on potato dextrose agar (PDA) and for GC-MS analysis, *M. phaseolina* was grown in potato dextrose broth (PDB) (HiMedia, India) at 28°C. The antagonistic bacterial strain was isolated from jute seed as an endophytic bacterium as described by Coombs and Franco [22, 23] and identified as *B. contaminans* by 16S rRNA. Bacterial subculture was maintained on Tryptone Soy Agar (TSA) (HiMedia, India).

Jute seedlings were used to assess plant growth promotion activity of *B. contaminans* NZ. Fresh seeds of a jute variety (*Corchorus olitorius* var. O-4) were collected from BJRI. All tests were performed in triplicate if not mentioned otherwise.

### Whole genome sequencing of *B. contaminans* NZ

*B. contaminans* NZ was cultured overnight in Tryptic Soy Broth (TSB) medium and incubated in an incubator shaker at 37°C. The genomic DNA was extracted using the GenElute™ Bacterial Genomic DNA Kit (Sigma, Germany). A genomic library was constructed and employed for 300-bp paired-end whole-genome sequencing at the Genome Research Institute of North South University, Dhaka, Bangladesh using an IlluminaMiSeq platform (Illumina, San Diego, CA, USA) according to the manufacturer's instructions. The generated raw reads (10xcoverage) were assembled using SPAdes version 3.11 [24]. The generated scaffolds were mapped and ordered using ABACAS–a reference-based assembler [25], considering *Burkholderia contaminans* CH1 as the reference genome [GenBank accession no. AP018357 to AP018360 (four entries)] [26].

**Genome sequence analysis and annotation.** For structural annotation of the genome Rapid Annotations using Subsystems Technology (RAST) server [27] was used (http://rast.nmpdr.org) along with SEED database for the functional annotation of predicted gene models. SEED also provides subsystem (collection of functionally related protein families) and derived

FIGfams (protein families) which represent the core of RAST annotation engine [28]. Subsequently, genome annotation through PIFAR annotation tool was used for the identification of bacterial genetic factors involved in plant host interaction [29]. Moreover, antiSMASH bacterial version 5.0 (https://antismash.secondarymetabolites.org/#!/start) [30] analysis was carried out to identify the biosynthetic gene clusters for secondary metabolites of *B. contaminans* NZ.

**Examining the presence of proteins fostering virulence and pathogenicity.** A total of 147 *Burkholderia* specific proteins reported for pathogenicity and virulence were collected from Virulence Factor Database (VFDB) (http://www.mgc.ac.cn/VFs/main.htm) [31] and investigated for the presence of virulence and pathogenicity related proteins in *B. contaminans* NZ with the help of BLASTp considering the e-value of at least 1e-10 and sequence identity, not less than 70%.

**Data deposition.** This Whole Genome Shotgun project has been deposited in DDBJ/ENA/GenBank under the accession QRBC00000000. The version described in this paper is version QRBC01000000.

### *In vivo* growth promotion study of *B. contaminans NZ* on jute

**Inoculation of jute seeds with the endophyte.** At first *B. contaminans* NZ was cultured on TSA plates and then inoculated in TSB media and incubated at 37°C at a shaking speed of 180 rpm for 24–48 h. At the same time, scarification was done for jute seeds with sandpaper and the seeds surface was sterilized using 5% NaOCl. Sterilized seeds were dipped into *B. contaminans* NZ suspension. From McFarland standard, bacterial cells of $10^8$ CFU/ml were determined by a serial dilution method [32].

**In vivo pot experiment in hydroponic culture.** Jute seeds inoculated with a bacterial suspension were grown under controlled environmental conditions in a plant growth chamber (Weiss Technik India Private Limited) at 28°C, 70% relative humidity, and 16-h light/8-h dark cycle. The seeds were allowed to germinate for three days and later grown in a hydroponic culture system. 1.25 ml per litre of a modified Yoshida medium ($NH_4NO_3$ 91.4g, $K_2SO_4$ 71.4g, $NaH_2PO_4.2H_2O$ 40.3g, $CaCl_2.2H_2O$ 88.6g, and $MgSO_4.7H_2O$ 324.0g per litre) and micronutrients ($MnCl_2.4H_2O$ 1.5g, $(NH_4)_6Mo_7O_{24}.4H_2O$ 0.074g, $H_3BO_3$ 0.934g, $ZnSO_4.7H_2O$ 0.035 g, $CuSO_4 .5H_2O$ 0.31g, $FeCl_3 .6H_2O$ 7.70 g, citric acid 11.9 g and $H_2SO_4$50.0g per litre) [33] was added to the seedlings. pH of the solution was adjusted to 5.5 with NaOH. Thirty seeds were sown 1 cm deep in a cork sheet in separate pots. Jute plants were grown for 10 days in hydroponic culture and 5 plants were collected for the measurement of fresh and dry weights, shoot, and root lengths on 4, 7, and 10 days after transfer of the seedlings into the hydroponic solution. The experiment was repeated three times.

### Antifungal activity assay

**Dual culture assay.** *In vitro* bacteria-fungal dual-culture assays were established by both agar well diffusion and cross streak methods in 9 cm diameter Petri dish systems containing PDA medium [34]. A 5 mm plug taken from the plate of an actively growing fungal colony was inoculated on one side of the Petri dish. Fresh cells of *B. contaminans* NZ were either streaked in 3 cm length parallel lines on the other side of the fungal plug or 20 μl of the overnight bacterial liquid culture was introduced into a 6 mm diameter well punched aseptically with a pipette tip in an agar plate. Control treatments containing only the fungus were also set up. The plates were incubated at 28°C for 4 to 5 days. The experiment was set up for four replicates. After incubation, the diameters of the fungal colonies were scored and measured.

Antifungal activity of *B. contaminans* NZ was determined against all the studied phytopathogenic fungi and the toxicity was expressed as a percentage of growth inhibition (PGI) and

calculated according to the Zygadlo *et al.*[35] formula,

$$PGI\ (\%) = 100\ (GC - GT)/GC$$

where GC represents the average diameter of the fungus grown on PDA (control); GT represents the average diameter of the fungus co-cultivated on the PDA dish with the antagonistic bacterium. A paired t-test was used to check whether the percentage of growth inhibition was significant.

**Co-culture of *M. phaseolina* and *B. contaminans* NZ.**   For GC-MS analysis, *M. phaseolina* was cultured in 20 ml and 100 ml PDB media in two separate conical flasks. *B. contaminans* NZ was cultured in 80 ml and 100 ml of TSB media in two separate conical flasks.

On the second day of incubation, 20 ml *M. phaseolina* culture was mixed with 80 ml *B. contaminans* NZ to form the co-culture and put to incubation again. All the cultures were incubated at 28˚C for 5 days for volatile compound extraction and observed for any phenotypic changes in *M. phaseolina* culture in the presence of *B. contaminans* NZ.

For HPLC and LC-MS analysis, a separate 1000 ml culture of *B. contaminans* NZ was cultured under similar conditions described above.

## Extraction of organic compounds from culture media

On day 6 of single cultures of fungus, bacterium, and co-cultured media were filtered using Whatman filter paper and centrifuged at 4000 rpm for 7 minutes to remove cell debris. Supernatants were separated into individual conical flasks. 1:1 ethyl acetate (v/v) was added to each flask, shaken, and kept for 4 hours. The organic phases were separated and dried using a rotary evaporator. Finally, the extracts were dissolved in 2 ml of ethyl acetate. A separate extraction (for HPLC and LC-MS analysis) of *B. contaminans* NZ culture media was also conducted using n-hexane instead of ethyl acetate and shaken for 24 hrs. The extracted fraction was finally dissolved in methanol.

**GC-MS analysis of ethyl acetate extract.**   Ethyl acetate extract of extracellular components was subjected to GC-MS analysis using Shimadzu GCMS-QP2010 Ultra; (Japan) mass detector connected with a capillary column of Rxi-5ms, 30 m long, 0.25 mm i.d.0.25 μm film thickness. 1μL of the sample was injected with splitless mode. Helium was used as the carrier gas with the flow rate was set at 1.0 ml/min. The column oven temperature was programmed from 40˚C (1 min hold) at a rate of 10˚C/min to 300˚C (10 min hold). Injector temperature was maintained at 250˚C. Temperatures of the mass interface and ion source were 250˚C and 200˚C respectively, and detector voltage was 0.4 KV. The analysis was carried out in the EI (electron impact) mode with 70e V of ionization energy. The analysis was performed in full scan mode, ranging from m/z 50 to 400. The compounds were detected after analyzing the mass spectrum of each component using the NIST11 library.

**HPLC and LC-MS analysis of methanolic extract.**   Analytical HPLC Dionex Ultimate 3000 with a C18 column (Nucleodur, 250x4.6 mm, particle size 5μm, pore size 110Å, 100-5C18ec) was used to analyze the bacterial extract dissolved in methanol. After dissolution, the extract was filtered and 200 μL was injected into the column. A multi-step gradient system (solvent system: acetonitrile and water with 0.05% trifluoroacetic acid; flow rate: 1ml/min; temperature: 23˚C) was performed with an optimized protocol (0–10 min, 10% acetonitrile; 10–30 min, 10–80% acetonitrile; 30–40 min, 80% acetonitrile; 40–50 min, 80–0% acetonitrile; volume fraction). The wavelength at 225 nm was used to detect compounds eluted from the column. Various fractions were collected containing high peak intensity in the chromatogram and the fractions were dried at low temperature in a lyophilizer (LYOVAC GT 2) and dissolved in methanol to test for antifungal activity and followed by a re-run in HPLC to confirm the reproducibility of the peak. The purified active fraction was then subjected to LC-MS.

For Liquid Chromatography (LC), Agilent 1290 Infinity II instrument was used. C18 column (2.1 mm x 100 mm x 1.8 um) (Zorbax RRHD Eclipse) was used as the stationary phase and 50:50 ratio of water (with 0.1% formic acid) and acetonitrile was used as the mobile phase on isocratic mode. The column temperature and the mobile phase flow rate were kept at 30˚C and 0.2 mL/min throughout the program. The sample injection volume was 5 uL. For Mass Spectrometry (MS) analysis, an Agilent 6420 Triple Quad Mass Spectrometer System was used, equipped with an electrospray ionization source operating at positive mode and scanning from *m/z* 100 to 1000. Nitrogen was used as the nebulizer gas, with pressure and flow set at 45 psi and 11 L/min, respectively. The capillary voltage was maintained at 4000 V, and dry gas temperature at 350˚C. Mass Hunter software was used to control and analyze the data. The compound was identified based on its MS fragmentation and analyzed according to literature [36].

### Identification of homologous melanin pathway(s) in *M. phaseolina*

Fungi contain different pathways for melanin biosynthesis; among them two pathways namely 1,8-dihydroxynaphthalene (DHN)-melanin and L-3,4-dihydroxyphenylalanine (DOPA)-melanin pathways, are primarily found in the ascomycota group of fungi [37]. *M. phaseolina* proteins homologous to this pathway were searched using BLAST. Pathway proteins discovered through literature review and BLAST searches were tested against the *M. phaseolina* proteome. At least a 30% identity cut-off was taken to consider the homology between the target and query protein [38].

### Statistical analysis

The average shoot length, average root length and average height from the three repeated experiment of *in vivo* pot experiment was taken for statistical analysis. Data obtained were subjected to analysis of variance (ANOVA) test against the control. The results presented as average means, standard deviation (SD), and standard error (SE) were determined by following the standard procedures.

## Results

### Pot tests for *in vivo* assay of plant growth promotion

*B. contaminans* NZ significantly increased the root and shoot lengths, and the average height of jute seedlings ($P<0.05$) (**Fig 1**, **S1 and S2** Tables). The maximum root length of 11.28 cm, was 116.9% greater than that of the control.

The effect on shoot length was also found to be significant; on average, it was 50.7% longer than that of the control jute plants. Fresh weight of 4-, 7-, and 10-day old jute seedlings increased by 141%, 122%, and 41%, respectively, compared to the controls ($P<0.05$).

In the presence of *B. contaminans* NZ, the dry weight of jute plants on an average, increased by 49.8% in all three replicates for each of the 4-, 7-, and 10-day measurements.

### Whole genome data analysis

A summary of the whole genome annotation of *B. contaminans* NZ sequenced using Illumina MiSeq technology is presented in **S3 Table**. Functional analysis of *B. contaminans* NZ using the RAST server (Rapid Annotation using Subsystem Technology, http://rast.nmpdr.org), antiSMASH, and PIFAR (plant–bacteria information factor resource) revealed numerous genes and gene clusters (**Table 1**, **S4 Table**), which are reported to be exclusively associated with plant growth promotion.

As observed in other PGPR *Burkholderia* strains [39], the *nifHDK* operon required for nitrogen fixation was detected in *B. contaminans* NZ, along with the 1-aminocyclopropane-1-carboxylate (ACC) deaminase coding sequence, gene coding for iron(III) ABC transporter

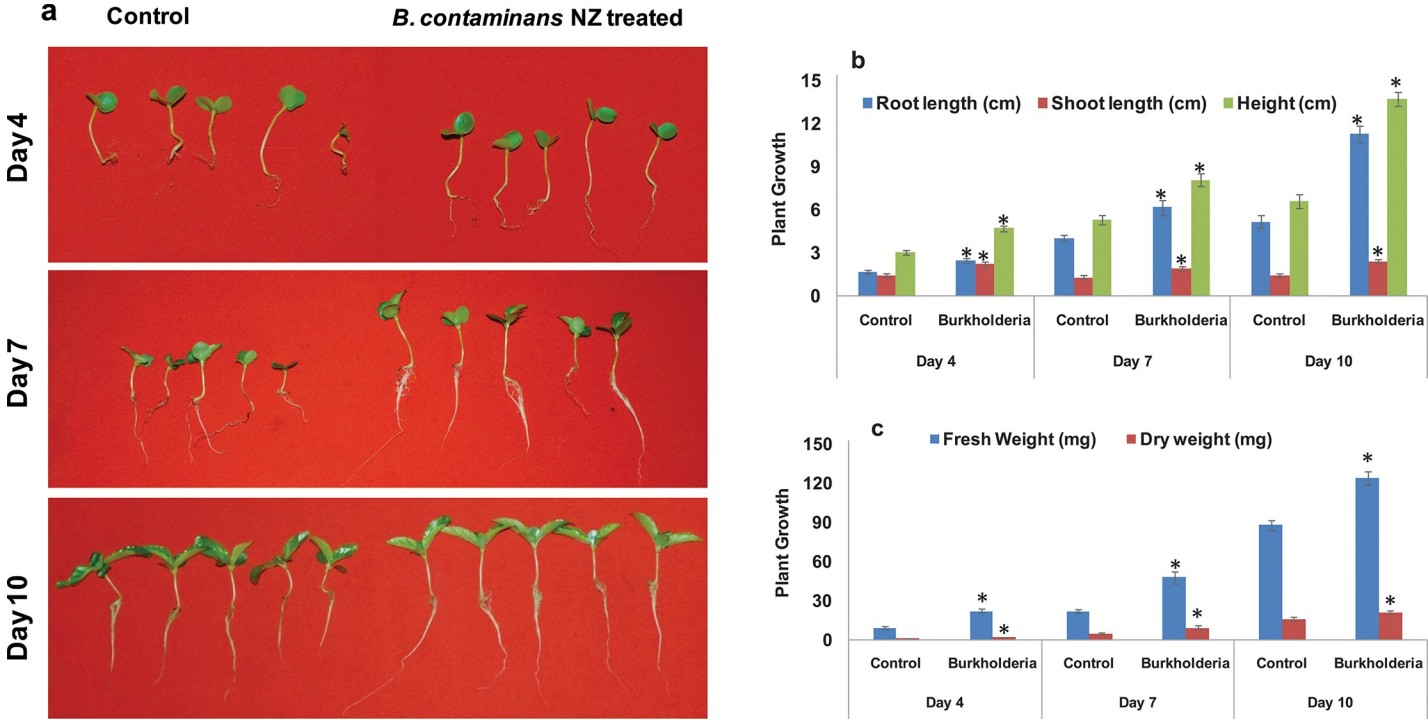

**Fig 1. *In vivo* effect of plant growth promoting endophytic bacteria *B. contaminans* NZ.** (a) Bacterial effect significantly influences the increase in root and shoot lengths of jute seedlings when compared with the control jute plants at 4, 7, and 10 days. Comparisons of (b) shoot and root lengths, plant heights, and (c) fresh and dry weights of *B. contaminans* NZ-treated and control jute seeds. Error bars represent the standard error of the mean of the replicates. Shoot and root lengths are represented in cm, and fresh and dry weights in mg. Asterisk (*) denotes the difference between control and endophyte-treated samples at a significance level of $P \leq 0.05$, as determined by ANOVA test. Values are mean(s) ± SD.

substrate-binding protein, phytoene synthase gene, the coding sequence for phosphotransferase system (PTS), and major facilitator superfamily (MFS) transporter genes. *In silico* analysis using RAST also revealed the presence of genes for indoleacetamide hydrolase involved in IAA

**Table 1. List of genes involved in plant growth promotion activity detected from RAST, antiSMASH, PIFAR analysis of the *B. contaminans* NZ whole genome.**

| Properties | Name | Biosynthetic genes |
|---|---|---|
| Phytohormone production and stress alleviation | Management of ethylene stress | ACC (1-Aminocyclopropane-1-carboxylic acid) deaminase |
| | Production of IAA (Indole Acetic Acid) | Indole-3-glycerate phosphate synthase |
| | | Indole pyruvate oxidoreductase |
| | | Indole acetamide hydroxylate |
| | | Tryptophan synthase |
| | | Nitrile hydratase |
| Phosphate solubilization | pyrroloquinoline quinone gene | *pqq* |
| | Enolase | *eno* |
| Nitrogen fixation | nif gene cluster | *nifHDK, nifQ,* |
| | Others | *nodT, nir, nor, nolO* |
| Antibiotic biosynthesis | | |
| | Pyrrolnitrin | *prnA-prnD* |
| Siderophore Biosynthesis | Polychelin | *pchR* |
| | Ferric siderophore transport | *pchD-pchA* |
| | ABC-type siderophore export system | *feoB* |
| | Siderophore pyoverdine | *pvd* |

production, which is a plant hormone associated with plant growth [40], and the *eno* gene, which assists in phosphate solubilization. Genome analysis also indicated the presence of pyrroloquinoline quinone synthase and glucose dehydrogenase, which are implicated in gluconic acid and 2-ketogluconic acid production, and required for mineral phosphate solubilization [41]. The annotated genome of *B. contaminans* NZ also contains 30 siderophore-related genes associated with ferric siderophore transport, ABC-type siderophore export system, arthrobactin, siderophore pyoverdine, and an intact siderophore pyochelin biosynthesis *pch* gene cluster. From the PIFAR data, in *B. contaminans* NZ, the biosynthesis of the major siderophore pyochelin is apparently associated with five genes (PMID: 22261733) distributed over five different regions in the bacterial genome.

While examining the core biosynthesis genes, genetic loci related to antibiotic production were also found in *B. contaminans* NZ, as presented in **Table 1**. Moreover, genes for pyrrolnitrin, an antifungal secondary metabolite produced by certain Bcc and several other gram-negative bacteria, were also identified. This metabolite inhibits the growth of a wide range of fungi and pathogenic bacteria [42, 43].

## Most virulence genes absent in *B. contaminans* NZ

Various virulence factors reported in pathogenic bacteria and other plant endophytic strains of *Burkholderia* (*B. contaminans* CH1, *B. contaminans* MS14) were screened for in the *B. contaminans* NZ genome, which failed in identifying major virulence-related genes, such as the biosynthesis genes for cable pili, toxoflavin, cepacian, and O-antigen (O-Ag) biosynthetic cluster (**Table 2**). This differs from pathogenic strains that contain many virulence genes. However, few virulent genes in *B. contaminans* NZ are present as incomplete gene clusters.

## Antifungal activity assay

In dual culture, *B. contaminans* NZ substantially inhibited the growth of six plant fungal pathogens, namely *Nigrospora sphaerica*, *Xylaria* spp., *Aspergillus fumigatus*, *Aspergillus niger*, *Penicillium oxalicum*, and *Rhizoctonia solani* (**Fig 2**).

**Table 2. List of major virulence related genes present or absent in *B. contaminans* NZ.**

| Feature name | Relevant gene/gene cluster | Status |
|---|---|---|
| Actin based intracellular motility | *BimA* | Absent |
| Adherence | *BoaA* | Absent |
|  | *BoaB* | Absent |
|  | Type VI pilin system | Incomplete cluster (3 out of 11 genes) |
| Anti-phagocytosis (O antigens) | Capsule I gene cluster | Absent |
| Secretion system | Bsa T3SS cluster | Absent |
|  | T6SS-1 cluster | Absent |
| Signaling | *Cdp* | Present |
|  | Quorum sensing | Incomplete cluster (2 out of 8 genes) |
| Cepacian | *bceA-bceK, bceN- bceT* | Absent |
| Cable pilin gene | *cblA* | Absent |
| Toxoflavin | *toxR, toxA-toxE* | Absent |
| 2-heptyl-3-hydroxy-4(1H)-quinolone | *pqsA-pqsE* | Absent |
| Hydrogencyanide | *hcnA-hcnC* | Absent |
| Cu2+ and Zn2+ containing periplasmic SOD | *apaH-reG* | Present |
| Zinc metalloprotease | *Zmp* | Present |

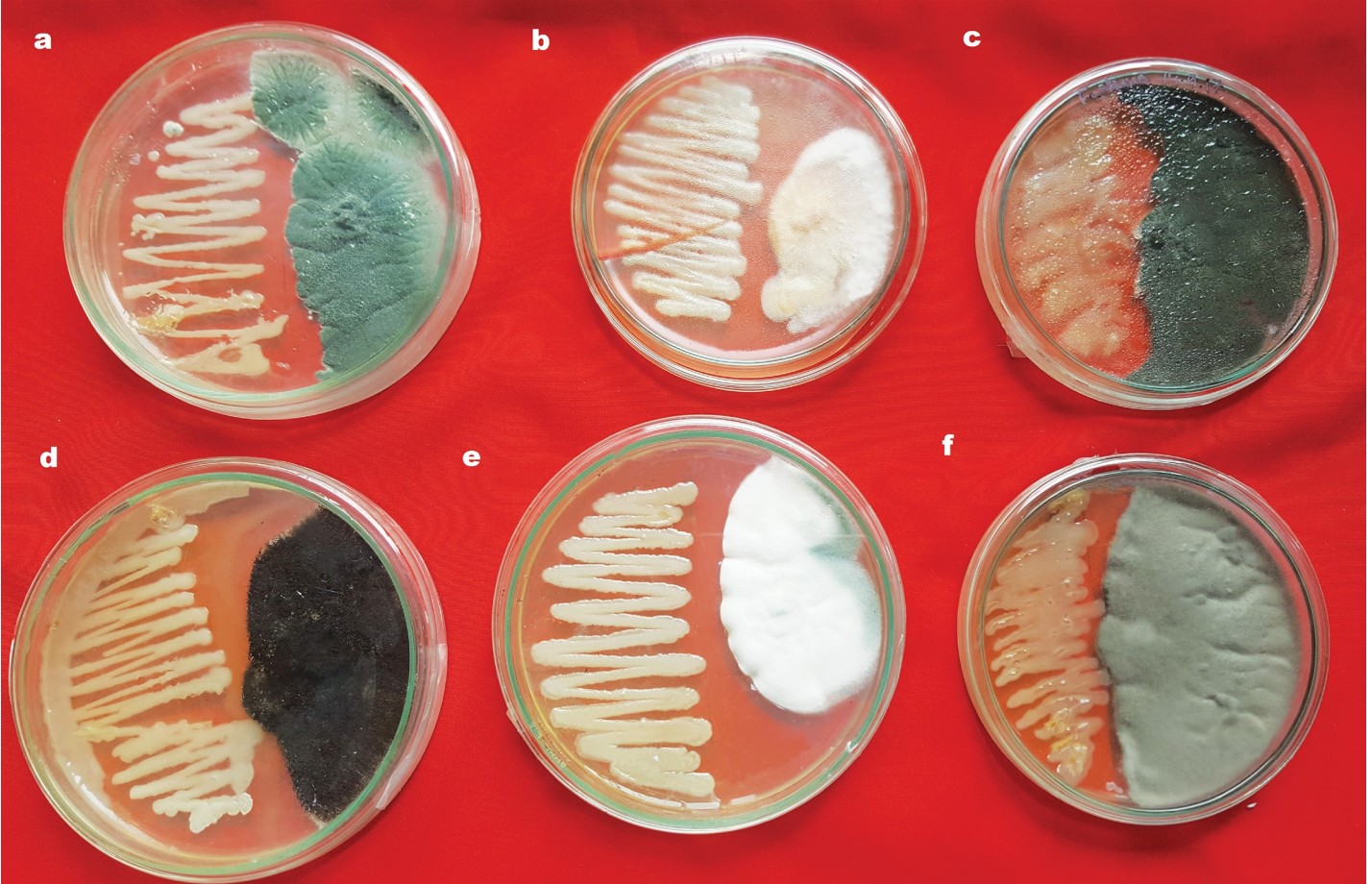

**Fig 2. Antagonistic properties of *B. contaminans* NZ against six other phytopathogenic fungi.** *(a) Nigrospora sphaerica* (b) *Xylaria* spp.(c) *Aspergillus fumigatus* (d) *Aspergillus niger* (e) *Penicillium oxalicum*, and(f) *Rhizoctonia solani.*

In dual-culture assay, *B. contaminans* NZ demonstrated significant growth inhibitory activity, which ranged from 44% to 58% than that of the control (P<0.05), when co-cultured with all the tested fungi. The growth inhibition rates observed for each fungal species are shown in Table 3.

## Growth suppression and chromogenic aberration in *B. contaminans* NZ challenged *M. phaseolina*

*M. phaseolina* inoculated in liquid culture was initially pale in color, which assumes a characteristic black color within two to three days that perpetuates until day 5. However, a

**Table 3. Growth inhibition induced by *B. contaminans* NZ on different plant pathogenic fungi.**

| Fungal species | Growth inhibition (%) (Mean ± SD) |
|---|---|
| *Xylaria* spp. | 57.71 ± 1.88 |
| *Aspergillus fumigatus* | 45.94 ± 1.45 |
| *Aspergillus niger* | 54.38 ± 4.10 |
| *Penicillium oxalicum* | 55.64 ± 2.96 |
| *M. phaseolina* | 49.04 ± 3.41 |
| *Nigrospora sphaerica* | 52.99± 3.67 |
| *Rhizoctonia solani* | 44.31 ± 4.36 |

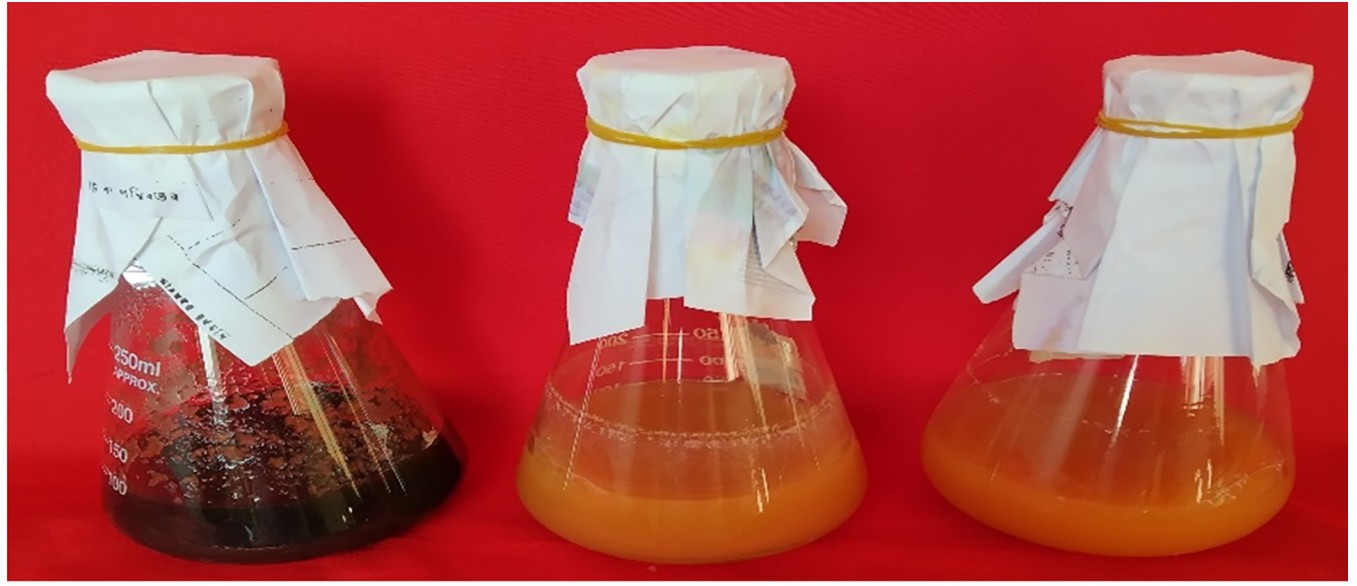

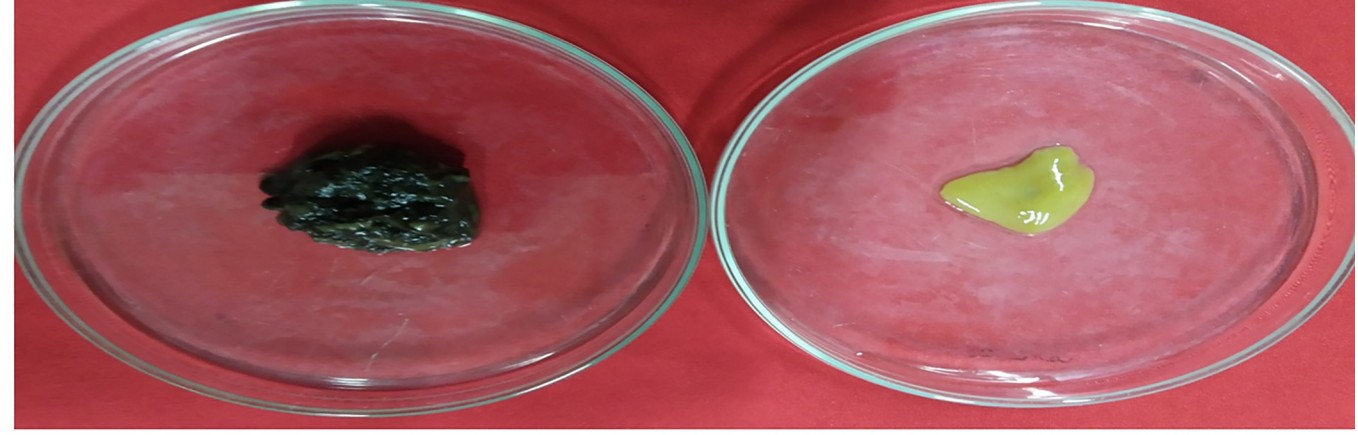

**Fig 3. Chromogenic aberration in *B. contaminans* NZ challenged *M. phaseolina*.** (A) *M. phaseolina*, *B. contaminans* NZ, and their co-cultures on day 5. (B) *M. phaseolina* without *B. contaminans* NZ retains the black color (left) compared to its *B. contaminans* NZ challenged counterpart (right). Inhibition of *M. phaseolina* growth and its pigmentation in the presence of *B. contaminans* NZ is evident compared to the culture containing only *M. phaseolina*.

*M. phaseolina* culture challenged by *B. contaminans* NZ deviates from this usual behavior and retains the initial pale color up to day 5. Furthermore, *M. phaseolina* growth when subjected to co-culture with the bacterium remained practically static from day 2. **Fig 3** shows the phenotypic changes in fungal growth from days 2 to 5.

## Chemical analysis of bio-active compounds using GC-MS

GC coupled with MS was used to identify the organic compounds produced by the fungus, bacteria, and their co-culture, and to identify the chemical nature of the bioactive compounds responsible for inhibiting fungal growth. Three sets of secondary metabolites obtained from (i) *B. contaminans* NZ, (ii) *M. phaseolina*, and (iii) *B. contaminans* NZ -*M. phaseolina* dual culture aided in identifying more than 72 compounds. Among them, 9,10-dihydro-12'-hydroxy-2'-methyl-5'-(phenylmethyl) ergotaman 3',6',18-trione; 2,3-dihydro-3,5-dihydroxy-6-methyl-4H-pyran-4-one; 1-(1,6-dioxooctadecyl)- pyrrolidine; 9-octadecenamide; 3-trifluoroacetoxy-pentadecane;2-hexyldecanol; and catechol, reportedly exhibit various biological activities, including tyrosinase inhibition, antifungal, and antimicrobial effects (**Table 4**). For example, catechol acts as a suicide inhibitor of enzyme tyrosinase [44], which forms melanin in fungi. Furthermore, 9, 10-dihydro-12'-hydroxy-2'-methyl-5'-(phenylmethyl) ergotaman 3', 6', 18-trione acts as an antimicrobial and anti-inflammatory agent [45]. A partial list of compounds and their corresponding functions are provided in **Table 4,** and peaks from GC-MS analyses of *M. phaseolina* (M), *B. contaminans* NZ (B), and *M. phaseolina* and *B. contaminans* NZ co-culture (C) are shown in **Fig 4**.

**Identification of pyrrolnitrin in methanolic extract of *B. contaminans* NZ culture.** The chromatogram of RP-HPLC analysis of the methanolic crude extract showed various peaks (**Fig 5A**). RP-HPLC fractions were collected and their activity tested against *M. phaseolina*, wherein the fraction obtained at a retention time of 33.2 min exhibited antifungal activity (**Fig 5B**). The compound with a molecular ion peak of 257.1 (*m/z*) was identified as pyrrolnitrin via LC-MS analysis, after comparing with standard pyrrolnitrin (**Fig 5C**).

**Table 4. List of potential compounds with their biological activities and retention times found in GC-MS analysis of *B. contaminans* NZ (B), *M. phaseolina* (M), and their co-culture (C).**

| Sl. No. | Compound Name | Culture Name | Retention Time (min) | Biological Activity | Reference (s) |
|---|---|---|---|---|---|
| 1 | 2,5-Dimethyl pyrazine | B, C | 6.28 | Fungistatic activity | [46] |
| 2 | 2-Methoxy- phenol | C | 9.30 | Antioxidant, Cytotoxic | [47] |
| 3 | 5-Methyl-furfural | M, C | 7.1 | Antimicrobial, Antioxidant | [48] |
| 4 | 2-Undecanethiol, 2-methyl | M, C, B | 9.01/12.6 | Antimicrobial activity | [49] |
| 5 | 2,3-Dihydro-3,5-dihydroxy-6-methyl-4H-pyran-4-one | B, C | 10.4 | Antifungal, Oxidative stress inducer | [50, 51] |
| 6 | Catechol | C | 11.4 | Antifungal activity, Suicide substrate inhibitors of tyrosinase | [44, 52, 53] |
| 7 | 1,4-Benzenediol / Hydroquinone | B | 11.42 | Cytotoxic activity, Protein kinase inhibition | [54] |
| 8 | 2-(1,1-Dimethylethyl)-4,6-dinitro-phenol | B | 12.8 | Herbicide | [55] |
| 9 | Decahydro-1,4-dimethoxy naphthalene | B | 14.6 | Antimicrobial activity | [56] |
| 10 | 2-Hexyldecanol | C | 16.06 | Suppresses melanin synthesis | [57] |
| 11 | 3-trifluoro acetoxypentadecane | C | 16.6 | Antimicrobial activity | [58] |
| 12 | Methoxyacetic acid, 3-tridecyl ester | M, B | 16.6/17.8 | Cytotoxicity | [59] |
| 13 | Pyrrolo[1,2-a]pyrazine-1,4-dione | M, B | 19.6/19.9 | Antioxidant agent, Antifungal activity, Antibiotic activity | [60–62] |
| 14 | 3,6-Diisopropylpiperazin-2,5-dione | B | 20.1 | Antimalarial activity | [63] |
| 15 | 1-(1,6-Dioxooctadecyl)- pyrrolidine | B, C | 20.23/21.2 | Antioxidant activity | [64] |
| 16 | n-Hexadecanoic acid | B, M, C | 21.04 | Antibacterial, Antifungal, Cytotoxicity, | [65] |
| 17 | Hexadecamethylheptasiloxane | B, C | 23.8 | Antifungal | [66] |
| 18 | 9-Octadecenamide | B, C | 24.8 | Antimicrobial, Anti-inflammatory | [58] |
| 19 | 9,10-Dihydro-12'-hydroxy-2'-methyl-5'-(phenylmethyl)-, (5' alpha,10 alpha) - ergotaman-'3',6',18-trione | B, C | 24.9/25.1 | Antimicrobial, Anti-inflammatory, Alpha-amaylase inhibitory activity | [45] |
| 20 | Di-n-octyl phthalate | B, M | 25.9/ 26.35 | Antifungal and antioxidant | [67, 68] |

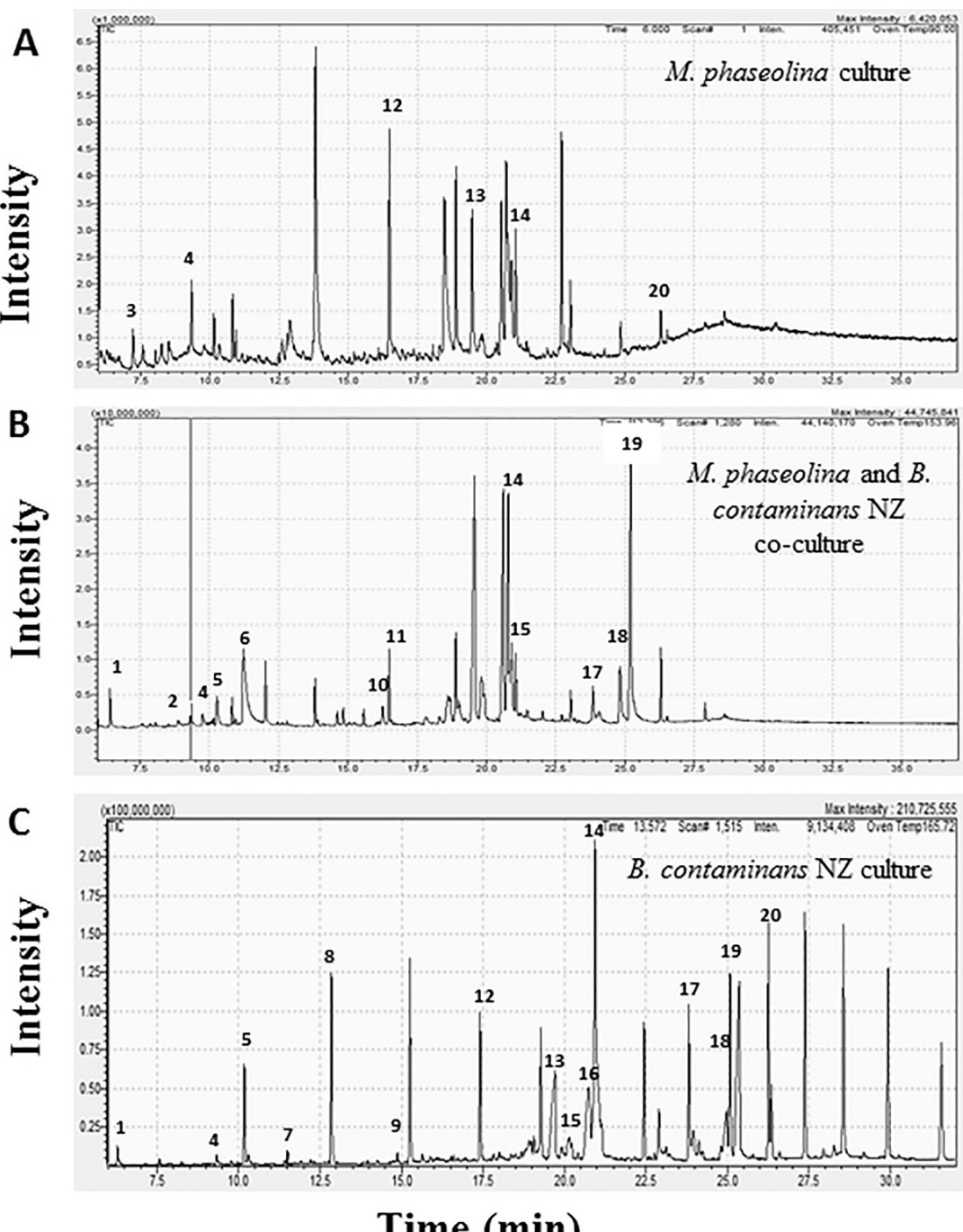

**Fig 4. Peaks from GC-MS analyses with potent volatile compounds.** (A) *M. phaseolina*, (B) *M. phaseolina* and *B. contaminans* NZ co-culture, and (C) *B. contaminans* NZ. The number on different peaks corresponds to the serial number of the compounds in **Table 4**.

## Identification of putative melanin pathways in *M. phaseolina* through homology-based search

After an extensive literature review of different melanin pathways active in fungi, polyketide synthase, tetrahydroxy naphthalene reductase, trihydroxy naphthalene reductase, scytalone

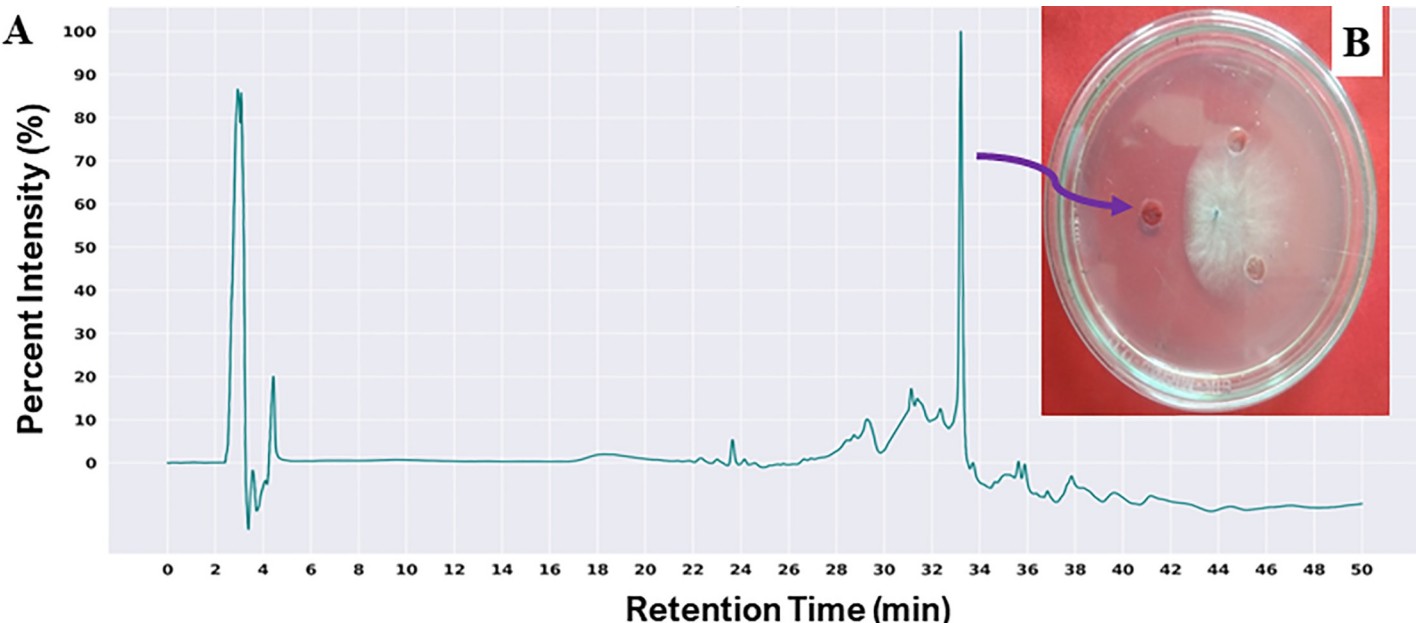

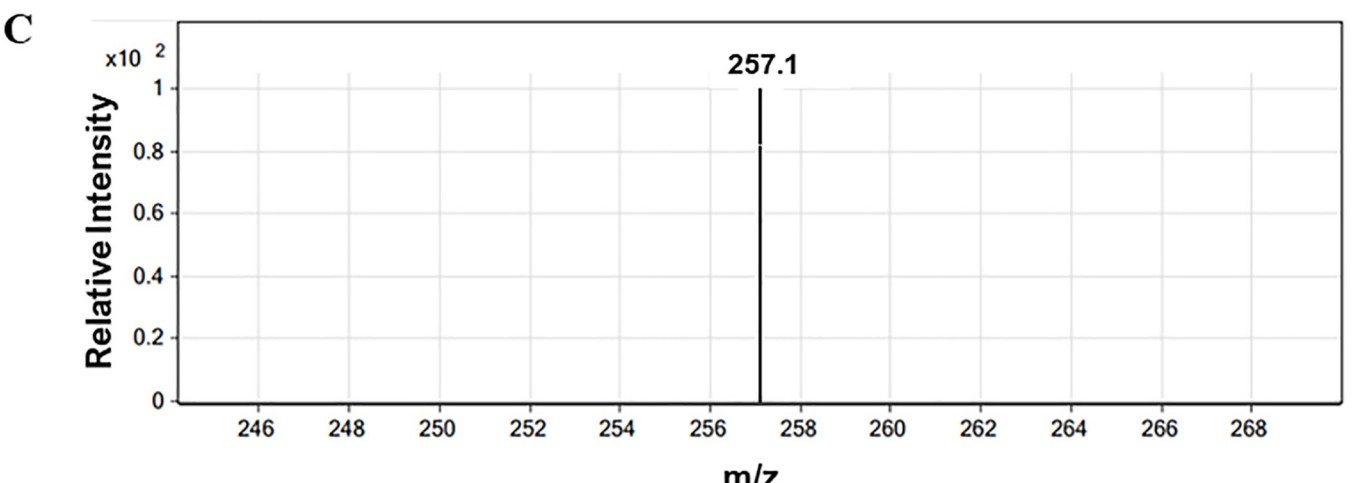

**Fig 5. Identification of bioactive compounds in methanolic extract of *B. contaminans* NZ culture.** (A) HPLC chromatograms of the active fraction (peak at 33.2 min retention time) upon reinjection. (B) Peak which inhibits the mycelial progression of *M. phaseolina* towards the active fraction when applied on the well, marked by a violet arrow. (C) LC-MS chromatogram of active peak identified to be pyrrolnitrin with a mass of 257.1 Da.

dehydratase, and tyrosinase proteins were selected, and their corresponding homologs in *M. phaseolina* were identified through a BLAST search. The details are presented in **Table 5**. In addition, putative pathways proposed in **S1 Fig** presents the order in which the reactions occur during melanin synthesis in *M. phaseolina*.

## Discussion

Since chemical control and most cultural practices are not effective representative tools that limit the pathogenic fungal growth and their distribution [69], the use of biocontrol agents has become a promising alternative in agriculture. Biocontrol using plant growth-promoting rhizobacteria (PGPR) is a potentially attractive and efficient disease management approach, as it

**Table 5. Homologous proteins of melanin pathways found in *M. phaseolina*.**

| Query Accession | Query Name | Subject Accession | Subject Name | Percent Identity |
|---|---|---|---|---|
| K2S0W9 | Betaketoacyl synthase | D7RJP3 | Polyketide synthase | 31.25 |
| K2RKI0 | Short-chain dehydrogenase/ reductase SDR | Q12634 | Tetrahydroxy naphthalene reductase | 75 |
| K2RKI0 | Short-chain dehydrogenase/ reductase SDR | W3XC32 | Trihydroxy | 75.5 |
| | | | Naphthalene | |
| | | | reductase | |
| K2R7Y4 | Scytalone dehydratase | W3XEE6 | Scytalone | 82.7 |
| | | | dehydratase | |
| K2R777 | Tyrosinase | U7PL38 | Tyrosinase | 31.132 |

promotes plant growth, enhances tolerance to abiotic stress [70], suppresses pathogens locally, and induces systemic resistance (ISR) against a broad range of crop diseases [71, 72].

Several studies conducted on endophytes have emphasized their ability in promoting plant growth and their additive/synergistic effects on plant growth and protection. In general, such growth-promoting rhizospheric bacteria belong to the following species: *Alcaligenes*, *Arthrobacteria*, *Azospirillum*, *Azotobacter*, *Bacillus*, *Burkholderia*, *Enterobacter*, *Klebsiella*, and *Pseudomonas* [8, 73]. The endophytic strain *B. contaminans* NZ isolated from jute has also been characterized *in vivo* for its potential plant growth-promoting (PGP) traits, which significantly increased both yield and biomass when jute seedlings were inoculated with the bacterium, and notably reduced the development of disease symptoms caused by *M. phaseolina*.

As *B. contaminans* NZ exhibited growth promotion and apparent protection against the phytopathogen *M. phaseolina*, its whole genome was sequenced to determine the characteristic genetic features. Genome annotation of *B. contaminans* NZ assisted in identifying genes related to phytohormone production, stress alleviation, phosphate solubilization, nitrogen fixation, antibiotic biosynthesis, and siderophore biosynthesis. *In vitro* experiments of the endophyte also displayed significant production of the above-listed compounds (**S5 Table**, **S2 Fig**).

Certain *Burkholderia* species demonstrate clinical associations and plant-pathogenic traits [74, 75]; therefore, genes related to virulence and pathogenicity were also investigated in the *B. contaminans* NZ genome. Several bacterial species from the *B. cepacia* complex (Bcc) are opportunistic pathogens [76]. *B. cenocepacia* strains expressing both cable (Cbl) pili and 22-KDa adhesin proteins have been reported to bind strongly to cytokeratin 13 (CK13), which efficiently infects the host cell by invading the squamous epithelium [77]. However, the *cblA* (giant cable pili) gene was absent in the symbiotic and legume-nodulating species [78]. *B. contaminans* NZ also lacks the cable pili biosynthesis gene cluster, indicating their inability to attach to the host cell, which is required to initiate infection. Furthermore, VgrG-5, which is a *Burkholderia* type VI secretion system 5-associated protein, required for complete mammalian virulence [79], is also absent in *B. contaminans* NZ genome. Virulence-associated protein sequences (for chemotaxis, attachment, type 3 and type 6 secretion systems; T3SS and T6SS) and typically described hallmark features representing the true and opportunistic pathogenic *Burkholderia* strains are also absent in *B. contaminans* NZ. Cystic fibrosis (CF)-related O-antigen of lipopolysaccharides associated with transmitting infections in CF patients [80] and zinc metalloproteases that may be involved in the overall virulence of several Bcc strains [81] are also absent. Among the virulence-associated proteins of *Burkholderia*, only the *sodC* gene encoding a $Cu^{2+}$ and $Zn^{2+}$ containing periplasmic superoxide dismutase that contributes in intracellular survival, which indicates self-protection ability in CF patients, is present in the genome [82]. *Burkholderia* species commonly produce plant-toxic secondary metabolites, polysaccharides, and other toxins such as rice grain rot and wilt causal agent, toxoflavin; exo-

polysaccharide toxin, cepacian; hydrogen cyanide (HCN); and 2-heptyl-3-hydroxy-4(1H)-quinolone [80]. These pathogenic genes were absent in this jute endophytic bacterium.

The *B. contaminans* NZ genome also contained antibiotic-biosynthetic genes. Thus, to gather further evidence, *M. phaseolina* and several other phytopathogenic fungi, namely, *Xylaria* spp., *Aspergillus fumigatus*, *Aspergillus niger*, *Penicillium oxalicum*, *Nigrospora sphaerica*, and *Rhizoctonia solani* were co-cultured with *B. contaminans* NZ, where all fungi tested were suppressed with varying degrees of inhibition (**Fig 2, Table 3**), confirming its antifungal activity.

Growth inhibition of *M. phaseolina* by *B. contaminans* NZ caused marked morphological changes and chromogenic aberration in the phytofungus [13]. Such morphological changes in the cell membrane have also been reported for *B. cepacia*-mediated inhibition of *F. solani* and *C. dematium* [83]. *Burkholderia* CF66I noticeably alters *R. solani* hyphae with multiple branches and swelling [84]. In previous reports, co-culturing *M. phaseolina* and *B. contaminans* NZ in solid media caused the fungus to lose its characteristic black color, restricted its growth, and attenuated its infectivity [13]. The present study also observed a similar deviation in pigmentation and growth repression in liquid co-culture (**Fig 3**).

Various reports have demonstrated that secondary metabolites produced by certain bacteria can influence fungal growth [85, 86]. Therefore, secondary metabolites were analyzed to explain chromogenic aberration and growth suppression of *M. phaseolina* when co-cultured with *B. contaminans* NZ. GC-MS analysis of ethyl acetate extracts of culture supernatant revealed over 72 compounds, among which 20 compounds are biologically important; 2,5-dimethyl pyrazine, 2,3-dihydro-3,5-dihydroxy-6-methyl-4H-pyran-4-one, hexadecamethylheptasiloxane, 9-octadecenamide, and 9,10-dihydro-12'- hydroxy-2'-methyl-5'- (phenylmethyl)-, (5' alpha,10 alpha)- ergotaman-3',6',18- trione compounds were discovered in both *B. contaminans* NZ and its co-cultures, but were absent in the *M. phaseolina* extract. 2,5-dimethylpyrazine, 2,3-dihydro-3,5-dihydroxy-6-methyl-4H-pyran-4-one, and hexadecamethylheptasiloxane exhibits antifungal activities [46, 50, 51, 87]. In addition, 9-octadecenamide and 9,10-dihydro-12'- hydroxy-2'-methyl-5'- (phenylmethyl)-, (5' alpha,10 alpha)- ergotaman-3',6',18- trione reportedly exhibits antimicrobial properties [58, 88]. The absence of these compounds in *M. phaseolina* extract implies that they are most likely produced only by *B. contaminans* NZ, thereby inhibiting the pathogen in the liquid media. 2-hexyldecanol, which suppresses melanin biosynthesis [57], and catechol, which demonstrates antifungal properties [52], are exclusively produced in the co-culture. Catechol, is also known as a suicide substrate for enzyme tyrosinase [53], which is involved in the DOPA-melanin biosynthesis pathway [37]. These compounds apparently produced only by *B. contaminans* NZ during co-culture condition are expected to play a role in the chromogenic aberration and growth suppression of *M. phaseolina*. n-Hexadecanoic acid possesses antifungal, antibacterial, and cytotoxic activities [65] and 2-undecanethiol, 2-methyl possesses antimicrobial activity [49]. Both compounds are ubiquitously present in *B. contaminans* NZ, *M. phaseolina*, and their co-cultures. Both organisms appear to be armed with antagonistic compounds, ready to inhibit each other in an attempt to avail the available resources.

*Burkholderia* spp. reportedly produces potent antifungal compound pyrrolnitrin [89]. This compound exhibits inhibitory effects by obstructing the synthesis of vital biomolecules (DNA, RNA, and protein), uncoupling oxidative phosphorylation, impeding mitotic division, and inhibiting several biological mechanisms. The genome of *B. contaminans* NZ also contained a gene cluster for pyrrolnitrin synthesis (**Table 2**). Although the GC-MS data revealed the presence of several compounds that implied possible modes for *M. phaseolina* growth suppression, pyrrolnitrin in the secondary metabolite profile was absent. Therefore, the extraction process was altered by substituting the extraction solvent with n-hexane instead of ethyl acetate to

identify pyrrolnitrin, which is in agreement with earlier reports that employed a similar method [43]. Thereafter rest of the method remained similar except that the extract was dissolved in methanol instead of ethyl acetate. RP-HPLC was performed to purify the active compound(s) to detect the appreciable suppression of fungal growth by the crude extract. Only one fraction exhibited considerable inhibitory activity against *M. phaseolina*, and its mass was identified using LC-MS analysis (257.1 Da), which was identical to the standard pyrrolnitrin (molecular weight: 257.07 g/mol).

A previous iTRAQ proteomic analysis of *M. phaseolina* challenged with *Burkholderia* showed the downregulation of beta-ketoacyl synthase, scytalone dehydratase, tyrosinase, enzymes of the DHN-melanin, and DOPA-melanin pathways [13]. Catechol present in the co-culture of *B. contaminans* NZ and *M. phaseolina* can explain the downregulation of tyrosinase. Kojic acid [5-hydroxy-2-(hydroxymethyl)-4H-pyran-4-one] is a prominent tyrosinase inhibitor [37], and 2,3-dihydro-3,5-dihydroxy-6-methyl-4H-pyran-4-one, identified by GC-MS analysis, is an oxidized form of kojic acid and is expected to act identical to its reduced counterpart. This derivatization could have occurred during sample processing for GC-MS analysis.

Further investigation of the *M. phaseolina* enzymes involved in melanin synthesis, similar to those reported earlier in other fungi, led to the identification of SDR–short chain dehydrogenase/reductase, which is similar to both trihydroxy naphthalene reductase [90] and tetrahydroxy naphthalene reductase [91], which are enzymes of the DHN-melanin pathway [92]. Tyrosinase [92, 93] performs multiple catalysis in the DOPA-melanin pathway; it converts phenylalanine to tyrosine, tyrosine to DOPA, and DOPA to DOPA quinone, which is eventually converted to melanin further downstream. However, both DOPA-melanin and DHN-melanin pathways are yet to be fully elucidated. This reduces the scope of finding other homologous pathway proteins in *M. phaseolina*. The percent identity of beta-ketoacyl synthase and tyrosinase was lower (~31%) than that of tetrahydroxy naphthalene reductase, trihydroxy naphthalene reductase, and scytalone dehydratase (~75% or more). This was possibly due to only few reports that describe the two genes with precise genomic features. Based on this evidence, a tentative melanin pathway was proposed for *M. phaseolina* in this study (**S1 Fig**).

## Conclusion

Whole genome analysis and secondary metabolite characterization have acknowledged *B. contaminans* NZ as a good biocontrol agent, in addition to its role in plant growth promotion. It suppresses the growth of multiple fungi, especially the phytopathogen *M. phaseolina*, by using catechol, pyrrolnitrin, and other antimicrobial agents. Compounds identified in the extracts of *B. contaminans* NZ or the co-culture of *B. contaminans* NZ and *M. phaseolina* contain compounds that inhibit melanin biosynthesis, which possibly contributes to the observed chromogenic aberration and growth suppression of the fungi. *B. contaminans* NZ can be established as a bio-control agent by conducting toxicity tests to ensure its safety, followed by field experiments to determine its efficacy under different environmental conditions. Hence, further studies are needed to optimize the formulation and application methods of *B. contaminans* NZ to fully maximize its potential as an effective agent in controlling *M. phaseolina*.

## Supporting information

**S1 Fig. Proposed melanin pathways in *M. phaseolina* based on homology search.**
(TIF)

**S2 Fig.** Qualitative assay for (a) siderophore, (b) ACC deaminase, and (c) nitrogen of *B. contaminans* NZ.
(TIF)

**S1 Table. Average root length, shoot length, and plant height of bacteria treated jute seedlings vs. untreated control in a pot experiment at 4, 7, and 10 days.** The data of three replicates per experiment are presented as means and standard deviations.
(DOCX)

**S2 Table. Fresh weight and dry weight of the bacteria treated jute seedlings vs. untreated control in a pot experiment at 4, 7, and 10 days.** The data of three replicates per experiment are presented as mean(s) and standard deviation(s).
(DOCX)

**S3 Table. Genomic features of *Burkholderia contaminans* NZ.**
(DOCX)

**S4 Table. Location of genes involved in plant growth promotion activity detected from RAST, antiSMASH, and PIFAR analysis of the whole genome of *B. contaminans* NZ.**
(DOCX)

**S5 Table. *In vitro* plant growth promotion attributes of *B. contaminans* NZ.**
(DOCX)

## Acknowledgments

The authors cordially thank NSU Genome Research Institute, North South University, Bangladesh for the whole genome sequencing and Dr. Abdul Baten, AgResearch Ltd. Christchurch, NZ for helping with the whole genome analysis and Enayet Hossain for collecting the GC-MS data. We also acknowledge the Bangladesh Council of Scientific and Industrial Research (BCSIR) for providing support to carry out the GC-MS.

## Author Contributions

**Conceptualization:** Nazia R. Zaman, Umar F. Chowdhury, Rifath N. Reza, Farhana T. Chowdhury, Mohammad Riazul Islam, Haseena Khan.

**Data curation:** Umar F. Chowdhury, Farhana T. Chowdhury, Mrinmoy Sarker, Muhammad M. Hossain, Md. Ahedul Akbor.

**Formal analysis:** Nazia R. Zaman, Rifath N. Reza, Farhana T. Chowdhury, Muhammad M. Hossain, Md. Ahedul Akbor, Al Amin, Haseena Khan.

**Funding acquisition:** Haseena Khan.

**Investigation:** Nazia R. Zaman, Umar F. Chowdhury, Rifath N. Reza, Mrinmoy Sarker, Al Amin, Mohammad Riazul Islam.

**Methodology:** Nazia R. Zaman, Umar F. Chowdhury, Mrinmoy Sarker, Muhammad M. Hossain, Md. Ahedul Akbor, Al Amin.

**Project administration:** Nazia R. Zaman, Mohammad Riazul Islam, Haseena Khan.

**Supervision:** Mohammad Riazul Islam, Haseena Khan.

**Validation:** Nazia R. Zaman.

**Writing – original draft:** Nazia R. Zaman, Umar F. Chowdhury, Farhana T. Chowdhury.

**Writing – review & editing:** Mohammad Riazul Islam, Haseena Khan.

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
