## [Decision Letter · Decision Letter 0]

30 Jun 2021

PONE-D-21-18655

Plant growth promoting endophyte Burkholderia contaminans NZ antagonizes phytopathogen Macrophomina phaseolina through multiple modes of action

PLOS ONE

Dear Dr. Islam,

Thank you for submitting your manuscript to PLOS ONE. After careful consideration, we feel that it has merit but does not fully meet PLOS ONE’s publication criteria as it currently stands. Therefore, we invite you to submit a revised version of the manuscript that addresses the points raised during the review process.

We look forward to receiving your revised manuscript.

Kind regards,

Abhay K. Pandey

Academic Editor

PLOS ONE

Journal Requirements:

"We gratefully acknowledge Higher Education Quality Enhancement Project (HEQEP) (Grant number: CP-3250), a World Bank financed development project, for funding the research."

"We gratefully acknowledge Higher Education Quality Enhancement Project (HEQEP) (Grant number: CP-3250), a World Bank financed development project, for funding the research. The funders had no role in study design, data collection and analysis, decision to publish, or preparation of the manuscript."

6. We note that Figures 1,2,and 3 in your submission contain copyrighted images. All PLOS content is published under the Creative Commons Attribution License (CC BY 4.0), which means that the manuscript, images, and Supporting Information files will be freely available online, and any third party is permitted to access, download, copy, distribute, and use these materials in any way, even commercially, with proper attribution. For more information, see our copyright guidelines: http://journals.plos.org/plosone/s/licenses-and-copyright.

a. You may seek permission from the original copyright holder of Figures 1,2,and 3 to publish the content specifically under the CC BY 4.0 license. 

Additional Editor Comments (if provided):

The English language must be addressed with the help of a native speaker or English language editing service. Furthermore, the manuscript lacks detailed experimental procedure; at least three replications and one repetition of experiment are required, failing which MS will be rejected. Discussion must focus the critical analysis of literature and should be condensed. Conclusion summarizes the findings of research.

Reviewers' comments:

Reviewer's Responses to Questions

**Comments to the Author**

1. Is the manuscript technically sound, and do the data support the conclusions?

Reviewer #1: Yes

Reviewer #2: Yes

2. Has the statistical analysis been performed appropriately and rigorously? 

Reviewer #1: Yes

Reviewer #2: Yes

3. Have the authors made all data underlying the findings in their manuscript fully available?

Reviewer #1: Yes

Reviewer #2: Yes

4. Is the manuscript presented in an intelligible fashion and written in standard English?

Reviewer #1: Yes

Reviewer #2: Yes

5. Review Comments to the Author

Reviewer #1: On the basis of what the authors reported in the manuscript, the present study was aimed to "Plant growth promoting endophyte Burkholderia contaminants NZ antagonizes phytopathogen Macrophomina phaseolina through multiple modes of action".

To gain these aims, the authors developed series of experiments devoted to re-culturing, identify pathogen and growth promoting bacterial isolates using GC-MS and whole genome sequencing and the information in results is clear and concise but need to be checked again as there are few grammatical error and used a few long sentences that should be summarized.

Materials and methods should be more attractive if the sequencing and analysis be placed before the in-vivo and in-vitro studies; over all the manuscript is written well and have valuable information for the readers. The whole manuscript should be proofread for English language editing and grammatical errors.

Discussion section of the manuscript is should be improved by using a few latest references.

Reviewer #2: 1. Objective of the study should be more clearly defined.

2. Replicates used in the experiments are very few, so my question is why the authors did not repeat the experiments. If they did, then it requires proper elaboration.

and How will authors justify the findings and their reproducibility?

3. Discussion is very very long, it should be précised.

4. Conclusion section is lacking impressiveness in its expression. I will suggest authors to put an extra effort in improving it.

5. There is also an suggestion about the title, I think it should be more specific as it is very generalized in its current form.

6. PLOS authors have the option to publish the peer review history of their article (what does this mean?). If published, this will include your full peer review and any attached files.

Reviewer #1: **Yes: **Raees Ahmed

Reviewer #2: No

---

## [Author Response · Author response to Decision Letter 0]

29 Jul 2021

Responses to the Editor and Reviewers’ comments

The format has been changed according to PLOS ONE’s style.

We would like to address the funding information as stated below:

This project is funded by Higher Education Quality Enhancement Project (HEQEP) (Grant number: CP-3250), a World Bank financed development project. The funders had no role in study design, data collection and analysis, decision to publish, or preparation of the manuscript."

There are no ethical or legal restrictions to sharing our data publicly. Data Availability statement has been corrected to ‘The authors have no objection to make the data set underlying the results described in our manuscript to be fully available.’

In the revised cover letter the accession no of the bacterial whole genome sequence (This Whole Genome Shotgun project has been deposited at DDBJ/ENA/GenBank under the accession QRBC00000000) and data availability statement have been included.

An ORCID iD was generated and validated in Editorial Manager

The results have been included in the Supporting Information (S2 Fig and S5 Table)

6. We note that Figures 1, 2, and 3 in your submission contain copyrighted images. All PLOS content is published under the Creative Commons Attribution License (CC BY 4.0), which means that the manuscript, images, and Supporting Information files will be freely available online, and any third party is permitted to access, download, copy, distribute, and use these materials in any way, even commercially, with proper attribution. For more information, see our copyright guidelines: http://journals.plos.org/plosone/s/licenses-and-copyright.

There is no chance of copyright issue for Figures 1, 2 and 3. They have been produced in this study and have not been used in any other article or report.

Additional Editor Comments (if provided):

The English language must be addressed with the help of a native speaker or English language editing service. Furthermore, the manuscript lacks detailed experimental procedure; at least three replications and one repetition of experiment are required, failing which MS will be rejected. Discussion must focus the critical analysis of literature and should be condensed. Conclusion summarizes the findings of research.

Reviewers' comments:

Reviewer's Responses to Questions

Comments to the Author

1. Is the manuscript technically sound, and do the data support the conclusions?

Reviewer #1: Yes

Reviewer #2: Yes

2. Has the statistical analysis been performed appropriately and rigorously?

Reviewer #1: Yes

Reviewer #2: Yes

3. Have the authors made all data underlying the findings in their manuscript fully available?

Reviewer #1: Yes

Reviewer #2: Yes

4. Is the manuscript presented in an intelligible fashion and written in standard English?

Reviewer #1: Yes

Reviewer #2: Yes

5. Review Comments to the Author

Reviewer #1: 

On the basis of what the authors reported in the manuscript, the present study was aimed to "Plant growth promoting endophyte Burkholderia contaminants NZ antagonizes phytopathogen Macrophomina phaseolina through multiple modes of action".

To gain these aims, the authors developed series of experiments devoted to re-culturing, identify pathogen and growth promoting bacterial isolates using GC-MS and whole genome sequencing and the information in results is clear and concise but need to be checked again as there are few grammatical error and used a few long sentences that should be summarized.

Materials and methods should be more attractive if the sequencing and analysis be placed before the in-vivo and in-vitro studies; over all the manuscript is written well and have valuable information for the readers. The whole manuscript should be proofread for English language editing and grammatical errors.

Discussion section of the manuscript is should be improved by using a few latest references.

Changes has been made according to the suggestions.

Reviewer #2: 

1. Objective of the study should be more clearly defined.

The objective has been clearly defined.

2. Replicates used in the experiments are very few, so my question is why the authors did not repeat the experiments. If they did, then it requires proper elaboration and How will authors justify the findings and their reproducibility?

All the experiments were done with three replications as stated in the Materials and Method section (lines 119 and 172) and all tests were performed in triplicate if not mentioned otherwise.

3. Discussion is very long, it should be précised.

Discussion has been modified accordingly.

4. Conclusion section is lacking impressiveness in its expression. I will suggest authors to put an extra effort in improving it.

This has been modified accordingly.

5. There is also an suggestion about the title, I think it should be more specific as it is very generalized in its current form.

The title has been edited as suggested.

6. PLOS authors have the option to publish the peer review history of their article (what does this mean?). If published, this will include your full peer review and any attached files.

Do you want your identity to be public for this peer review? For information about this choice, including consent withdrawal, please see our Privacy Policy.

Reviewer #1: Yes: Raees Ahmed

Reviewer #2: No

Reviewer Recommendation Term: 

Major Revision

1. Is the manuscript technically sound, and do the data support the conclusions? 

Yes

2. Has the statistical analysis been performed appropriately and rigorously? 

Yes

3. Have the authors made all data underlying the findings in their manuscript fully available?

Yes

4. Is the manuscript presented in an intelligible fashion and written in standard English?

Yes

6. Review Comments to the Author

The data itself is good, but the manuscript need to be improved in terms of writing. There are so many small mistakes that can easily be fixed by just reading it through a few more times. Here I listed a portion of the problems, but I recommend the authors to go over carefully for each sentence and craft it again. Also, I didn't like the red background for some figures. Although I won't say this is mandatory, I think red is probably not the best background color for those figures.

Overall, this paper presents solid data sets and I recommend it to be published after major revisions.

Abstract

Line 40- Secondary metabolites, catechols and ergotaman (that has been found through what?)

They were found through GC-MS analysis as mentioned in line 35.

Introduction

Line 58- be specific of what 'They' indicates (ex: the beneficial effects of endophytes)

Revised accordingly

Line 61-63

biocontrol activity including competition for iron, nutrient and space, production of antibiotics, lytic enzymes and volatile compounds, and induction of systemic resistance.

Corrected accordingly 

Line 63-65

Many studies have emphasized the ability of these microorganisms for possible roles for promoting plant growth and protection through additive/synergistic effects.

Corrected accordingly

Line 66

B. subtilis [9], Pseudomonas parafulva, and Pantoea agglomerans are a few examples.

Not so sure about "recent developments". What developments?

Corrected accordingly

Line 68

Full scientific name for Jute

Throughout its life cycle,

Corrected accordingly

Line 81

human environment? Like inside of human body? If so,

"a widespread presence in water, soil, plants, and animals including human"

Corrected accordingly

Line 84 & 86

Space between (Bcc) and [17] / [18] and .

Corrected accordingly

Line 90

Don't start a line with "but"

Corrected accordingly

Line 94-96

I think this line can be written shorter and clearer.

Changed accordingly

Overall, introduction is good, but can be better by reorganizing and rewriting sentences.

Line 105

Delete "used in this study ... collection of"

Corrected accordingly

Materials and methods

Line 108-109 and throughout manuscript

Use full scientific names including authroties if possible.

These should be okay.

Line 117

Fix 'NZ' (italic)

Corrected accordingly

Line 123

At first,

TSB media? Full names should be mentioned when first introduced. Instead of that, full name was introduced in line 199

Corrected accordingly

Line 124

At the same time,

Corrected accordingly

Line 134

and MgSO4.7H2O

Corrected accordingly

Line 137

Thirty instead of 30

Corrected accordingly

Line 152

(10x coverage)

Corrected accordingly

Line 158

server [29]

Corrected accordingly

Line 198 and throughout manuscript

Authors should be consistant for number+unit (ul, degree, etc) combination. For example, I can see inconsistancies such as 20ml (no space) vs 20 ml (space between number+unit). Just go with one.

Corrected accordingly

Line 205-206

For HPLC and LC-MS analysis, a seperate 1000ml of B. contaminans NZ was cultured under similar conditions described above.

Corrected accordingly

Line 208

Delete 'together' since co-culture already contains the meaning.

Corrected accordingly

Line 209

filtered thorough 'what'?

Corrected accordingly

Line 213-215

Recommend to rephrase it. Hard to understand.

Revised accordingly

Line 220

One µL is awkward. What about one microliter?

Corrected accordingly

Line 221

Helium was used as the carrier gas with the flow rate set at 1.0 mL/min

Also, be comsistant with ml or mL (chose one)

Corrected accordingly

Line 224

respectively, and

Corrected accordingly

Line 235

temperature:

Corrected accordingly

Line 251-252

The capillary voltage was maintained at 4000 V, and dry gas temperature was set at 350°C.

Corrected accordingly

Line 256

biosynthesis; among those,

Corrected accordingly

Results

Table 1.

Adding chromosomal locations for the genes would be beneficial.

Has been added in S4 Table

Line 334

and (f) Rhizoctonia solani

Corrected accordingly

Table 3.

Xylaria sp. 

Nigrospora sphaerica 52.99 ± 3.67

Corrected accordingly

Line 403-404

, and their 404 corresponding homologues in M. phaseolina were identified through a BLAST search 

Corrected accordingly

Discussion

Line 416

Delete ,

Corrected accordingly

Line 440

Delete But and replace with 'however' or similar word.

Corrected accordingly

Line 458

The Burkholderia genome

Corrected accordingly

Line 459

M. phaseolina 

Corrected accordingly

Line 461

and Rhizoctonia solani 

Corrected accordingly

Line 481

Space issue

Corrected accordingly

Line 530

, and scytalone dehydratase 

Corrected accordingly

Line 547

downregulation

Corrected accordingly

---

## [Decision Letter · Decision Letter 1]

16 Aug 2021

PONE-D-21-18655R1

Plant growth promoting endophyte Burkholderia contaminans NZ antagonizes phytopathogen Macrophomina phaseolina through melanin synthesis and pyrrolnitrin inhibition

PLOS ONE

Dear Dr. Islam,

Thank you for submitting your manuscript to PLOS ONE. After careful consideration, we feel that it has merit but does not fully meet PLOS ONE’s publication criteria as it currently stands. Therefore, we invite you to submit a revised version of the manuscript that addresses the points raised during the review process.

We look forward to receiving your revised manuscript.

Kind regards,

Abhay K. Pandey

Academic Editor

PLOS ONE

Additional Editor Comments (if provided):

This manuscript is an improvement over the previously submitted version. Although all reviewers recommended publication of the MS, however, The MS still needs improvement in terms of clarity and quality. I recommended in my last review that the authors find someone fluent in scientific writing in English to correct the many grammar, terminology, and sentence structure errors in the manuscript. While, some effort was made to correct these errors, the current manuscript is still filled with errors that make the article very difficult to read. I have noted a few of these errors on the manuscript, but there are many errors per page that need to be addressed. The level of writing is not up to the standard required for publication in PLoS series journals. At this point, the authors should consider employing a professional editor experienced in scientific writing.

In abstract, background of research and aim of the study are missing, please revise abstract to make it more focused.

What is NZ in abstract, this is a wrong way of presentation please revise, do you mean Burkholderia contaminans, if it is, then it is not properly cited.

Page 2 lines 34 compounds or molecules not substances, and overall the MS need thorough English revision failing which MS will be not accepted. Overall, abstract is very poorly written, please revise it to make more focused.

Page 3 line 65 line 85 species not italic correct in whole MS

Line 77 space

Line 92 to 101 it should described objective of your research and why it needed and then what are you presenting in this paper.

Did you check pathogenicity of Macrophomina phaseolina before conducting experiments?

Line 120 it was not grown in shaker, you incubated in incubator shaker, how long old culture you used for DNa isolation

Line 157 spore density was adjusted ??? revise sentence

Line 172, the experiment was repeated three times, if it is the case, how you handled the data from repeated experiments it is not described in the statistical section. Also this section does not show you used original data or transformed data for analysis.

Petri dish, P should be capital

Line 193 35 and 36 why two reference, anyway this paper contains more references, more than we require for a research paper, please reduce the number of references upto 50.

Remove statistical part from each section and put it in the last of material and method in the single section describing the details how u analyzed the data as I mentioned.

Line 259 ascomycota group of fungi

Line 272 (P<0.05)

Table1 I think name of genes should be italic, follow in all tables

Mention CD, MSS and F values in table 3.

Coulumn one Fungal Species

Line 342 full name Macrophomina phaseolina should start follow in whole in whole MS same trend

Table 4 can be given as supplementary table

Line 441 strains

Conclusion should be short and easily understandable

Some of the references not follow the journal trends.

Reviewers' comments:

Reviewer's Responses to Questions

**Comments to the Author**

1. If the authors have adequately addressed your comments raised in a previous round of review and you feel that this manuscript is now acceptable for publication, you may indicate that here to bypass the “Comments to the Author” section, enter your conflict of interest statement in the “Confidential to Editor” section, and submit your "Accept" recommendation.

Reviewer #1: All comments have been addressed

Reviewer #2: All comments have been addressed

Reviewer #3: All comments have been addressed

2. Is the manuscript technically sound, and do the data support the conclusions?

Reviewer #1: Yes

Reviewer #2: Yes

Reviewer #3: Yes

3. Has the statistical analysis been performed appropriately and rigorously? 

Reviewer #1: Yes

Reviewer #2: Yes

Reviewer #3: Yes

4. Have the authors made all data underlying the findings in their manuscript fully available?

Reviewer #1: Yes

Reviewer #2: Yes

Reviewer #3: Yes

5. Is the manuscript presented in an intelligible fashion and written in standard English?

Reviewer #1: Yes

Reviewer #2: Yes

Reviewer #3: Yes

6. Review Comments to the Author

Reviewer #1: I have received and checked critically the revisions from the author on the the manuscript with title "Plant growth promoting endophyte Burkholderia contaminans NZ antagonizes phytopathogen Macrophomina phaseolina through melanin synthesis and pyrrolnitrin inhibition" and found satisfactory.

On the light of this my recommendation is to accept the manuscript keeping in view the decision of other experts.

Reviewer #2: (No Response)

Reviewer #3: Authors corrected listed errors/problems, and this MS looks a lot better now.

Just a few recommendations:

Based on track change file

Line 81: There is a space between [14] & .

Line 125-126: What rpm?

7. PLOS authors have the option to publish the peer review history of their article (what does this mean?). If published, this will include your full peer review and any attached files.

Reviewer #1: **Yes: **Raees Ahmed

Reviewer #2: **Yes: **Adnan Akhter

Reviewer #3: No

---

## [Author Response · Author response to Decision Letter 1]

10 Sep 2021

Responses to the Editor and Reviewers’ comments

Additional Editor Comments (if provided):

This manuscript is an improvement over the previously submitted version. Although all reviewers recommended publication of the MS, however, The MS still needs improvement in terms of clarity and quality. I recommended in my last review that the authors find someone fluent in scientific writing in English to correct the many grammar, terminology, and sentence structure errors in the manuscript. While, some effort was made to correct these errors, the current manuscript is still filled with errors that make the article very difficult to read. I have noted a few of these errors on the manuscript, but there are many errors per page that need to be addressed. The level of writing is not up to the standard required for publication in PLoS series journals. At this point, the authors should consider employing a professional editor experienced in scientific writing.

The revised manuscript has been corrected by professional editing company.

In abstract, background of research and aim of the study are missing, please revise abstract to make it more focused.

Made changes according to the suggestions

What is NZ in abstract, this is a wrong way of presentation please revise, do you mean Burkholderia contaminans, if it is, then it is not properly cited.

Changed all instances of NZ to B. contaminans NZ 

Page 2 lines 34 compounds or molecules not substances, and overall the MS need thorough English revision failing which MS will be not accepted.

Changed according to the suggestions 

Overall, abstract is very poorly written, please revise it to make more focused.

Revised according to the suggestions

Page 3 line 65 line 85 species not italic correct in whole MS

Revised accordingly

Line 77 space

Revised accordingly

Line 92 to 101 it should described objective of your research and why it needed and then what are you presenting in this paper.

Revised accordingly

Did you check pathogenicity of Macrophomina phaseolina before conducting experiments?

We have checked the pathogenicity of Macrophomina phaseolina on jute seedlings and found jute seedlings dies after few days of infection, and turns brown. This result has been included in our recent paper by Zaman NR et al. (2020)

Ref: 

1. Zaman NR, Kumar B, Nasrin Z, Islam MR, Maiti TK, Khan H. (2020) Proteome Analyses Reveal Macrophomina phaseolina's Survival Tools When Challenged by Burkholderia contaminans NZ. ACS Omega. 5(3):1352-1362. doi:10.1021/acsomega.9b01870

Line 120 it was not grown in shaker, you incubated in incubator shaker, how long old culture you used for DNA isolation

The statement was corrected according to the suggestion. It is mentioned in the same line that the bacterium was grown overnight and DNA was isolated from that culture.

Line 172, the experiment was repeated three times, if it is the case, how you handled the data from repeated experiments it is not described in the statistical section. Also this section does not show you used original data or transformed data for analysis.

 The average shoot length, average root length and average height from the three repeated experiment was taken for statistical analysis shown in supplementary table 1.

Petri dish, P should be capital

Revised accordingly

Remove statistical part from each section and put it in the last of material and method in the single section describing the details how you analyzed the data as I mentioned.

Revised accordingly 

Line 259 ascomycota group of fungi

Revised accordingly

Line 272 (P<0.05)

Revised accordingly throughout the MS

Table1 I think name of genes should be italic, follow in all tables

Revised accordingly

Coulumn one Fungal Species

Revised accordingly

Line 342 full name Macrophomina phaseolina should start follow in whole in whole MS same trend

Revised accordingly

Table 4 can be given as supplementary table

Revised accordingly

Line 441 strains

Revised accordingly

Conclusion should be short and easily understandable

Changed accordingly

Some of the references not follow the journal trends.

Changed accordingly

---

## [Editor Report · Decision Letter 2]

14 Sep 2021

Plant growth promoting endophyte Burkholderia contaminans NZ antagonizes phytopathogen Macrophomina phaseolina through melanin synthesis and pyrrolnitrin inhibition

PONE-D-21-18655R2

Dear Dr. Islam,

We’re pleased to inform you that your manuscript has been judged scientifically suitable for publication and will be formally accepted for publication once it meets all outstanding technical requirements.

Kind regards,

Abhay K. Pandey

Academic Editor

PLOS ONE

Additional Editor Comments (optional):

authors addressed all comments
---

## [Editor Report · Acceptance letter]

22 Sep 2021

PONE-D-21-18655R2 

Plant growth promoting endophyte *Burkholderia contaminans* NZ antagonizes phytopathogen *Macrophomina phaseolina* through melanin synthesis and pyrrolnitrin inhibition 

Dear Dr. Islam:

I'm pleased to inform you that your manuscript has been deemed suitable for publication in PLOS ONE. Congratulations! Your manuscript is now with our production department. 

Kind regards, 

on behalf of

Dr. Abhay K. Pandey 

Academic Editor

PLOS ONE